# P-Glycoprotein (MDR1/ABCB1) Restricts Brain Accumulation of the Novel EGFR Inhibitor EAI045 and Oral Elacridar Coadministration Enhances Its Brain Accumulation and Oral Exposure

**DOI:** 10.3390/ph15091124

**Published:** 2022-09-08

**Authors:** Jing Wang, M. Merve Susam, Changpei Gan, Rolf W. Sparidans, Maria C. Lebre, Jos H. Beijnen, Alfred H. Schinkel

**Affiliations:** 1Division of Pharmacology, The Netherlands Cancer Institute, 1066 CX Amsterdam, The Netherlands; 2Division of Pharmacoepidemiology & Clinical Pharmacology, Department of Pharmaceutical Sciences, Faculty of Science, Utrecht University, 3584 CS Utrecht, The Netherlands; 3Division of Pharmacology, Department of Pharmaceutical Sciences, Faculty of Science, Utrecht University, 3584 CG Utrecht, The Netherlands; 4Department of Pharmacy & Pharmacology, The Netherlands Cancer Institute, Slotervaart Hospital, 1066 CX Amsterdam, The Netherlands

**Keywords:** EAI045, ABCB1, ABCG2, brain accumulation, elacridar, Oatp1a/1b, CYP3A4

## Abstract

EAI045 is a fourth-generation allosteric tyrosine kinase inhibitor (TKI) of the epidermal growth factor receptor (EGFR). It targets T790M and C797S EGFR mutants in the treatment of non-small cell lung cancer (NSCLC). EAI045 and cetuximab combined induce tumor regression in mouse models of EGFR-mutant lung cancer. We investigated the pharmacokinetic roles of the multidrug efflux and uptake transporters ABCB1 (P-gp), ABCG2 (BCRP), and OATP1A/1B, and of the drug-metabolizing enzyme CYP3A in plasma and tissue distribution of EAI045 and its metabolites, using genetically modified mouse models. In vitro, EAI045 was a good transport substrate of human ABCB1. In vivo, oral EAI045 (20 mg/kg) was rapidly absorbed. Relative to wild-type mice, EAI045 brain-to-plasma ratios were increased 3.9-fold in *Abcb1a/1b**^-/-^* and 4.8-fold in *Abcb1a/1b;Abcg2**^-/-^* mice. However, in single *Abcg2**^-/-^* mice they were unchanged. EAI045 oral availability was not markedly altered. Oral coadministration of elacridar, an ABCB1/ABCG2 inhibitor, increased the plasma AUC_0–30min_ and brain-to-plasma ratios of EAI045 by 4.0-fold and 5.4-fold, respectively, in wild-type mice. EAI045 glucuronide showed an increased plasma AUC_0–30min_ and a markedly decreased accumulation and tissue-to-plasma ratio in the small intestinal content when Abcb1a/1b and Abcg2 were absent. A large fraction of oral EAI045 was converted to its hydrolyzed metabolite PIA, but Abcb1a/1b, Abcg2, and Oatp1a/1b had little impact on PIA pharmacokinetics. Mouse Cyp3a knockout or transgenic human CYP3A4 overexpression did not significantly affect oral EAI045 pharmacokinetics. Our results show that blood–brain barrier ABCB1 can markedly limit EAI045 brain accumulation. Moreover, elacridar coadministration can effectively reverse this process.

## 1. Introduction

Lung cancer is the foremost cause of cancer death among women and men in many developed countries. For instance, for 2021, 235,760 new cases are expected and 131,880 persons are projected to die from lung cancer in the USA alone [1]. Non-small cell lung cancer (NSCLC) represents about 80% to 85% of all cases [2,3,4,5]. The past 20 years have shown tremendous progress in biological understanding and management of NSCLC. The development of the third-generation EGFR tyrosine kinase inhibitors (TKIs) represented a significant breakthrough in the treatment of patients with EGFR mutant, and especially T790M mutant, NSCLC. However, the tyrosine kinase domain C797S mutation is a leading mechanism of resistance to these third-generation irreversible EGFR inhibitors [6,7,8,9]. To overcome this C797S mutation, Jia et al. screened a library of approximately 2.5 million compounds against the L858R/T790M EGFR mutant kinase peptide [10]. Among these compounds, an EGFR allosteric inhibitor-1 (EAI001) was discovered. However, it only had modest potency against individual L858R and T790M mutants. With further modification and characterization, EGFR allosteric inhibitor-45 (EAI045) was found to have the most selective inhibition of the EFGR mutant relative to wild-type EGFR [10]. Moreover, EAI045 shows marked synergy with cetuximab, a therapeutic antibody that blocks dimerization of EGFR [11,12]. The combination of EAI045 with cetuximab is effective against lung cancer driven by EGFR(L858R/T790M) and by EGFR(L858R/T790M/C797S) in mouse models; EAI045 or cetuximab alone had a very modest effect in the L858R/T790M mutant mice, but marked tumor regressions were observed in these mice in the combination treatment [10,13].

Membrane transporters can play important roles in the pharmacokinetic, safety and efficacy profiles of drugs. Among these, ATP-binding cassette (ABC) and solute carrier (SLC) transporter superfamilies are widely expressed (e.g., brain, liver, small intestine, and kidney) and thought to have profound roles in drug disposition and in drug–drug and drug–food interactions [14,15]. Two important ABC efflux transporters, P-glycoprotein (P-gp, ABCB1, MDR1) and breast cancer resistance protein (BCRP, ABCG2), are expressed in the apical membrane of organs with absorptive and eliminatory functions, such as liver, intestine, and kidneys. They are also expressed in the endothelial cells of blood–tissue interfaces (such as the blood–retinal, blood–ovarian, blood–brain, and blood–testis barriers) [16,17,18,19,20,21,22,23,24]. This expression profile suggests that the central roles of ABCB1 and ABCG2 may be to prevent the entry of drugs or toxins into physiologically critical tissues and to mediate their elimination through renal, intestinal, and hepatobiliary excretory pathways [22,23,24].

Organic anion transporting polypeptides (rodents: Oatp, gene *Slco*; human: OATP, gene *SLCO*) are mostly uptake transporters. They can transport a wide range of endogenous and exogenous compounds, including anionic oligopeptides, steroid conjugates, bile salts, thyroid hormones, and many drugs, as well as other xenobiotics [25]. For instance, the OATP1A and 1B subfamilies are highly expressed in the liver, and appear to play an important role in the hepatic uptake and plasma clearance of many substrate drugs [26].

In the present study, we aimed to investigate the pharmacokinetic functions of ABCB1 and ABCG2 in vitro and in vivo, and the roles of OATP1A/1B in vivo, with respect to systemic exposure upon oral administration (in short: oral exposure) and tissue distribution of EAI045. We further examined the effect of co-administration of elacridar, an ABCB1 and ABCG2 inhibitor. Moreover, we aimed to determine to what extent EAI045 oral exposure and tissue accumulation is affected by multi-specific drug-metabolizing enzymes of the cytochrome P4503A (CYP3A) family.

## 2. Results

### 2.1. In Vitro Transport of EAI045

Transport of EAI045 (5 μM) across epithelia was tested using cell layers of Madin–Darby Canine Kidney (MDCK-II) cells and derived clones expressing hABCB1, hABCG2, or mAbcg2 cDNA. In the parental line, EAI045 showed modest transport in the apical direction (efflux ratio r = 1.7, Figure 1A). The ABCB1 inhibitor zosuquidar could completely inhibit this transport (r = 1.1, Figure 1B). EAI045 is, therefore, likely a good transport substrate of the endogenous canine ABCB1, expressed at low levels in the parental cell line [27]. In cells overexpressing hABCB1, there was clear transport of EAI045 in the apical direction (r = 5.5, Figure 1C). Zosuquidar extensively inhibited this transport (r = 1.3, Figure 1D). To evaluate the interaction between hABCG2 and mAbcg2 and EAI045, zosuquidar (5 μM) was added to inhibit any endogenous canine ABCB1. In cells overexpressing hABCG2, the BA transport of EAI045 was somewhat increased relative to the AB transport, resulting in an efflux ratio of 1.4 (Figure 1E). The addition of the ABCG2 inhibitor Ko143 reduced the efflux ratio to 1.1 (Figure 1F). mAbcg2-overexpressing MDCK-II cells displayed a slight apically directed transport of EAI045, r = 1.8 (Figure 1G), and Ko143 reduced this to an r of 1.4 (Figure 1H). EAI045 thus appears to be a good transport substrate of hABCB1, a moderate transport substrate of the endogenous canine ABCB1, whereas it is at best slightly transported by mouse Abcg2 or human ABCG2 in vitro.

### 2.2. Impact of ABCB1 and ABCG2 on Oral EAI045 Plasma and Tissue Distribution over 4 h

In view of the in vitro transport results, we assessed whether both transporters could affect EAI045 plasma pharmacokinetics (systemic exposure upon oral drug administration, or oral exposure in short) and tissue distribution in mice. A 4-h pilot experiment was performed in female wild-type (WT) and double-knockout *Abcb1a/1b;Abcg2^-/-^* mice, with oral administration of 20 mg/kg EAI045. As shown in Appendix A, EAI045 was very rapidly absorbed in all strains, with an apparent t_max_ around 5 min. The plasma exposure of EAI045 over 4 h (AUC_0–4h_) was not meaningfully altered between WT and *Abcb1a/1b;Abcg2^-/-^* mice (Appendix A). However, variation between mice was quite high, as often observed immediately after oral drug administration.

We further measured the brain, liver, kidney, spleen, lung, and small intestine with content (SIWC) concentrations of EAI045 4 h after oral administration. Whereas at t = 4 h the plasma concentrations between the two strains were similar, the brain concentrations and brain-to-plasma ratios in *Abcb1a/1b;Abcg2^-/-^* mice were dramatically higher than those in WT mice (25-fold and 12-fold, respectively, Appendix A). We also observed a modest but significantly higher liver concentration (2.8-fold) and liver-to-plasma ratio (1.6-fold) (Appendix A) in *Abcb1a/1b;Abcg2^-/-^* mice compared to WT mice. In contrast, the small intestine with content (SIWC) percentage of dose (SIWC%) and SIWC%-to-plasma ratio was reduced in *Abcb1a/1b;Abcg2^-/-^* mice (Appendix A). No significant differences were detected between the strains in kidney-, spleen-, and lung-to-plasma ratios (Appendix A). Collectively, these results show that Abcb1a/1b and/or Abcg2 can efficiently restrict the brain penetration of EAI045, and possibly they are involved in its hepatic and intestinal disposition.

EAI045 is hydrolyzed to a major metabolite, which is negatively charged under physiological conditions (PIA, Appendix A). We therefore studied the plasma and tissue concentration of PIA over 4 h after oral EAI045 administration at 20 mg/kg. There were no significant differences between the two mouse strains in plasma concentration (Appendix A), tissue concentrations, and tissue-to-plasma ratios (data not shown) of PIA. This suggests that Abcb1a/1b and/or Abcg2 play little or no role in limiting PIA availability and tissue distribution.

### 2.3. Effect of ABCB1 and ABCG2 on EAI045 Plasma and Tissue Distribution over 30 min

We subsequently analyzed the single and combined effects of Abcb1 and Abcg2 deficiencies on EAI045 plasma pharmacokinetics and tissue accumulation using male WT, *Abcg2^-/-^*, *Abcb1a/1b^-/-^*, and *Abcb1a/1b;Abcg2^-/-^* mice. As we wanted to assess tissue concentrations while plasma concentrations were still relatively high; we terminated this experiment at 30 min. As shown in Figure 2A,B, there were only limited differences in plasma EAI045 levels over 30 min (AUC_0–30min_) due to the absence of Abcb1a/1b or Abcg2. Loss of both Abcb1a/1b and Abcg2 resulted in a 1.7-fold increase in plasma AUC_0–30min_, which was not significant compared to WT mice, but also in a significant 2.1-fold increase relative to single *Abcb1a/1b^-/-^* mice (Figure 2A,B; Table 1). These results suggest that each of these transporters potentially has a minor limiting effect on the oral exposure of EAI045 in mice. As we had observed in the 4-h experiment, there was also high inter-individual variation for this 30-min experiment, especially during the first 10 min after administration in all mouse strains. This may be due to different rates of stomach emptying for this drug.

At 30 min, *Abcb1a/1b^-/-^* and *Abcb1a/1b;Abcg2^-/-^* mice showed increased brain concentrations compared to WT mice, by 3.5-fold and 7.2-fold, respectively (Figure 3A; Table 1). After correcting for the plasma concentrations, *Abcb1a/1b^-/-^* and *Abcb1a/1b;Abcg2^-/-^* mice showed significant increases in brain-to-plasma ratios, by 3.9-fold and 4.8-fold, respectively. However, no significant difference was seen between *Abcg2^-/-^* mice and WT mice (Figure 3B; Table 1). Moreover, in *Abcb1a/1b;Abcg2^-/-^* mice, the brain-to-plasma ratio was 4.8-fold increased (*p* < 0.001) relative to that in WT mice, and 5.2-fold increased (*p* < 0.001) compared to *Abcg2^-/-^* mice, but with no significant difference compared to *Abcb1a/1b^-/-^* mice (Figure 3B; Table 1). This shows that primarily Abcb1a/1b can markedly limit the brain penetration of EAI045. The testis distribution of EAI045 showed qualitatively similar behavior as its brain distribution. The testis-to-plasma ratios in *Abcb1a/1b^-/-^* and *Abcb1a/1b;Abcg2^-/-^* mice were significantly increased by 3.1-fold and 2.2-fold relative to WT mice, and by 4.5-fold and 3.1-fold compared to *Abcg2^-/-^* mice, respectively (Figure 3D; Table 1).

In contrast to brain and testis, the recovery of EAI045 (% of dose) in the SIWC was markedly reduced in the *Abcb1a/1b;Abcg2^-/-^* mice relative to the other mouse strains, especially when corrected for plasma exposure, but the single knockout strains showed no reduced recovery relative to WT mice (Figure 3E,F, Table 1). This observation suggests that both Abcb1a/1b and Abcg2 may affect enterohepatic circulation of EAI045 by performing intestinal excretion, and thus restricting net absorption across the intestinal epithelium. In this way, the absence of Abcb1a/1b and Abcg2 could reduce the amount of drug recovered from the intestinal content. At this early time point after EAI045 oral administration (30 min), a large amount of EAI045 was recovered from the intestinal lumen in WT mice (29.9%), and about half that amount in *Abcb1a/1b;Abcg2^-/-^* mice (14.2%) (Figure 3E, Table 1). However, at 4 h, there was only 1.7% left in WT intestinal lumen (Appendix A), suggesting that longer term EAI045 was not retained extensively in the intestinal lumen.

The other tissues we tested, including liver, kidney, spleen, and lung, demonstrated no meaningful changes in tissue concentrations or tissue-to-plasma ratios between the mouse strains, except for one, likely spurious, outlier for the liver-to-plasma ratio in *Abcb1a/1b^-/-^*, but not *Abcb1a/1b;Abcg2^-/-^* mice. Typical tissue-to-plasma ratios were about 5.7, 2.8, 1.1, and 2.6 for liver, kidney, spleen, and lung, respectively (Appendix A), much higher than the ratios observed for (WT) brain (0.10) and testis (0.11).

### 2.4. Effect of ABCB1 and ABCG2 on Plasma and Tissue Distribution of EAI045 Metabolites

After oral administration of EAI045, we also measured the PIA concentration in plasma and different tissues. Within 10 min PIA was more abundant in plasma than the parent drug in all strains, suggesting a relatively rapid conversion to this metabolite (Figure 2A,C). However, no meaningful differences in oral AUC_0–30min_ among the strains were observed (Figure 2C,D). The PIA/EAI045 AUC_0–30min_ ratios ranged between 270 and 480% and also did not show meaningful differences between the strains (data not shown). This suggests that any differences between the strains mainly reflected the higher EAI045 levels in the *Abcb1a/1b;Abcg2^-/-^* mice. For tissues, we found only a significant decrease in SIWC% (1.8-fold, *p* < 0.05) and SIWC%-to-plasma ratios (2.7-fold, *p* < 0.05) in *Abcb1a/1b;Abcg2^-/-^* mice compared to WT mice, again in line with the results found for EAI045. The absolute amount of PIA retrieved from SIWC at this time point was much smaller than that of EAI045 (~0.3% vs. ~30% of the EAI045 dose in WT mice); although the latter high percentage may well have represented yet unabsorbed EAI045 from the dosing. Furthermore, we observed no significant changes in tissue-to-plasma ratios among these strains for brain, testis, liver, kidney, spleen, and lung (data not shown). Brain penetration was extremely low, with the brain-to-plasma ratio at only 0.014 in WT mice, consistent with the negatively charged nature of PIA. Moreover, distribution of PIA to other tissues was generally much lower than that of the parent compound, with the possible exception of kidney. Altogether, the data suggest a very limited impact of the ABC transporters on PIA disposition, with the possible exception of intestinal excretion.

EAI045 itself is also conjugated to form EAI045-glucuronide; although the exact structure of the glucuronide metabolite is as yet unclear. We therefore analyzed the plasma concentration-time curve of EAI045 glucuronide after oral EAI045 administration at 20 mg/kg. For the EAI045 glucuronide, the plasma AUC_0–30min_ was significantly increased in *Abcb1a/1b;Abcg2^-/-^* mice, by 4.9-fold (*p* < 0.001) compared to WT mice, and by 3.2-fold and 2.7-fold compared to *Abcg2^-/-^* and *Abcb1a/1b^-/-^* mice (Figure 2E–F). Due to absence of an authentic standard, EAI045 glucuronide could only be semi-quantified, but this semi-quantification does allow relative concentration comparisons to be made within one tissue or matrix. For ease of comparison, we still used ng units for EAI045 glucuronide, but these should be considered arbitrary mass units. Owing to the higher plasma exposure of the glucuronide in *Abcb1a/1b;Abcg2^-/-^* mice; kidney and liver also had significantly higher glucuronide concentrations in this strain. Nonetheless, the tissue-to-plasma ratios did not reveal meaningful differences among the mouse strains (Appendix A). Interestingly, both SIWC%-to-plasma ratio and SIWC% accumulation were clearly reduced in *Abcb1a/1b;Abcg2^-/-^* mice compared to WT mice (4.4-fold and 5.5-fold), as well as to *Abcg2^-/-^* mice (2.6-fold and 2.5-fold) (Appendix A). The data suggest that Abcb1a/1b and Abcg2 are to some extent involved in the plasma clearance of EAI045 glucuronide, possibly through a relatively reduced hepatobiliary and/or direct intestinal excretion in *Abcb1a/1b;Abcg2^-/-^* mice. It should be noted, however, that the glucuronide response relative to EAI045 was low, 2.9% for kidney, 1.8% for liver, and 0.17% for SIWC (Appendix A), suggesting a very low amount of EAI045 glucuronide formed overall.

### 2.5. Co-Administration of Elacridar Boosts Brain and Testis Distribution of EAI045

As shown in Figure 3 and Table 1, there were marked differences between WT and *Abcb1a/1b;Abcg2^-/-^* mice in brain-to-plasma ratios, as well as testis and intestinal disposition. Considering the potential therapeutic benefit of increasing EAI045 brain penetration, and possibly also plasma exposure, we investigated to what extent elacridar, a dual ABCB1 and ABCG2 inhibitor, could increase the brain penetration and oral exposure of EAI045. We performed a 30-min pharmacokinetic experiment with an EAI045 dose of 20 mg/kg, and elacridar at 50 mg/kg. As elacridar plasma concentration peaks around 3–4 h after oral administration, elacridar was administered orally 3 h prior to oral EAI045 administration to male WT mice. Plasma and tissue EAI045 levels were determined 30 min later. In elacridar-treated WT (WT + Ela) mice, the EAI045 plasma AUC_0–30min_ was markedly increased compared to that in WT mice without elacridar treatment (4-fold, *p* < 0.001). Unexpectedly, in the presence of elacridar, the plasma AUC_0–30min_ of EAI045 was further increased significantly by 2.4-fold (*p* < 0.001) in WT + Ela mice compared to that in *Abcb1a/1b;Abcg2**^-/-^* mice (Figure 4A,B; Table 1). These results show that EAI045 oral plasma exposure in WT mice could be substantially enhanced by elacridar, but that likely other systems in addition to ABC transporters were affected by the elacridar treatment that can normally restrict EAI045 plasma exposure.

With respect to tissue distribution, elacridar coadministration dramatically enhanced the brain concentration of EAI045 in WT mice by 17-fold (*p* < 0.001), and brain-to-plasma ratios by 5.4-fold (*p* < 0.001). The brain-to-plasma ratio was thus similar to that observed in *Abcb1a/1b;Abcg2^-/-^* mice without elacridar (Figure 4C,D; Table 1). Coadministration of elacridar also enhanced the EAI045 testis distribution in WT mice, in a similar pattern as that seen for brain (Figure 4E,F; Table 1). In contrast, the liver-, kidney-, spleen-, and lung-to-plasma ratios of EAI045 were not noticeably affected by elacridar treatment in the WT mouse strain compared to WT and *Abcb1a/1b;Abcg2**^-/-^* mice without elacridar treatment (Appendix A).

Interestingly, elacridar treatment significantly decreased the percentage of dose present in the SIWC and the relative SIWC%-to-plasma ratio in WT + Ela mice by 7.5-fold and 30.8-fold, respectively, compared to WT mice without elacridar treatment. Moreover, these differences were also found between WT + Ela and *Abcb1a/1b;Abcg2^-/-^* mice, i.e., WT + Ela showed 3.5-fold and 8.6-fold decreases for SIWC% and SIWC%-to-plasma ratio, respectively, compared to *Abcb1a/1b;Abcg2^-/-^* mice (Appendix A). In conjunction with the disproportionately increased plasma levels of EAI045 in WT + Ela mice, this suggests that elacridar treatment can markedly improve the net intestinal uptake of EAI045. It therefore appears that elacridar can not only inhibit Abcb1a/1b and Abcg2 in the intestine and liver, but also some other system(s), perhaps another efflux transporter, that can restrict the net intestinal uptake of EAI045.

Collectively, these data indicate that oral elacridar coadministration can specifically and extensively inhibit the activity of mouse Abcb1 and Abcg2 in the blood–brain and blood–testis barriers, resulting in markedly enhanced EAI045 distribution to the brain and testis after oral administration. Furthermore, elacridar can possibly also inhibit EAI045 elimination system(s) different from Abcb1 and Abcg2, causing increased net drug absorption from the gut, leading to additionally increased plasma exposure and decreased small intestinal content retention of EAI045. Supporting an impact of elacridar primarily on the EAI045 absorption process, the main effect on enhancing EAI045 plasma concentrations was established within 5 min after oral administration, after which the relative EAI045 plasma clearance (assessed by semi-log plot) was similar between wild-type mice with and without elacridar (Figure 4A).

### 2.6. Impact of OATP1A/1B on Plasma and Tissue Distribution of EAI045 and Its Metabolites

OATP-mediated uptake into the liver can also impact the (oral) plasma exposure and clearance of a number of its drug substrates. Little to nothing is known about whether EAI045 interacts with OATP/SLCO uptake transporters. We therefore performed a pharmacokinetic experiment in male WT and *Oatp1a/1b^-/-^* mice, administering EAI045 (20 mg/kg) orally, and analyzing the plasma concentrations over 30 min, as well as the liver-to-plasma ratios at 30 min. Systemic exposure of EAI045 was not changed in *Oatp1a/1b^-/-^* mice compared to the WT mice (Figure 5A,B). We further observed no significant differences between WT and *Oatp1a/1b^-/-^* mice in tissue concentration or tissue-to-plasma ratios of EAI045 for brain, testis, spleen, lung, liver, and SIWC (Appendix A). This suggests that the mouse Oatp1a/1b transporters do not meaningfully affect EAI045 oral exposure and distribution to these tissues. We did observe a modestly, but significantly, higher kidney concentration and kidney-to-plasma ratio in *Oatp1a/1b^-/-^* mice relative to WT mice, but given the high experimental variation and the large number of tissues assessed, it is uncertain whether these differences are meaningful (Appendix A).

The potential effect of Oatp1a/1b on the pharmacokinetics of the, negatively charged, metabolite PIA was investigated also. Whereas the plasma concentrations of PIA were somewhat lower in *Oatp1a/1b^-/-^* than in WT mice, this difference was not statistically significant (Figure 5C,D). Loss of Oatp1a/1b further resulted in a 4.2-fold increase in kidney-to-plasma ratio, and a 3.3-fold increase in brain-to-plasma ratio (data not shown), suggesting that one or more of the Oatp1a/1b transporters might be involved in net efflux of PIA from these tissues. Perhaps more conventionally, we observed a 1.5-fold decrease in liver-to-plasma ratio of this metabolite, which might be consistent with reduced hepatic uptake of PIA due to Oatp1a/1b deficiency. No significant differences were found for testis, spleen, lung, and SIWC (data not shown).

For EAI045 glucuronide, the C_max_ and AUC_0–30min_ in plasma increased 15-fold and 16-fold in *Oatp1a/1b^-/-^* compared to WT mice (Figure 5E,F; Table 2). In *Oatp1a/1b^-/-^* mice the extrapolated plasma AUC_0–30min_ of the glucuronide was 7.86 h*ng/mL, so still very low compared to the EAI045 AUC_0–30min_ (136 h*ng/mL) in WT mice. Perhaps not surprisingly, loss of mOatp1a/1b resulted in a 22.6-fold decreased liver-to-plasma ratio and a 32.6-fold decreased liver accumulation (Figure 6D–F; Table 2) in *Oatp1a/1b^-/-^* mice compared to WT mice. This indicates that EAI045 glucuronide was efficiently taken up by mouse mOatp1a/1b into the liver. We also observed a significant decrease in SIWC%-to-plasma ratio (11.3-fold) and SIWC% accumulation (17.6-fold) in *Oatp1a/1b^-/-^* mice compared to WT mice (Figure 6G–I; Table 2). This probably reflects the dramatically reduced liver accumulation of the glucuronide, and likely subsequent biliary excretion into the intestinal lumen. In contrast, the kidney concentration was about 35-fold higher in *Oatp1a/1b^-/-^* mice, and after correction for the plasma concentration and AUC_0–30min_, the increases were still about 2.4-fold and 1.5-fold compared to WT mice (Figure 6A–C; Table 2). Possibly, Oatp1a/1b plays a minor role in the release of glucuronide from the kidney.

### 2.7. Assessment of the In Vivo Role of CYP3A in EAI045 Pharmacokinetics

To determine whether EAI045 and CYP3A interact in vivo, we carried out a 24 h pharmacokinetic pilot experiment using oral EAI045 at 20 mg/kg in female WT, Cyp3a knockout (*Cyp3a^-/-^*), and transgenic Cyp3aXAV mice (these are humanized *Cyp3a^-/-^* mice that express the human CYP3A4 in liver and small intestine). The plasma exposure and tissue distribution of EAI045 and PIA were measured. The t_max_ in all strains was around 5–30 min. The plasma exposure of EAI045 over 30 min (AUC_0–30min_), somewhat unexpectedly, was slightly, but significantly, lower in *Cyp3a^-/-^* mice than WT mice (1.6-fold). In contrast, relative to *Cyp3a^-/-^* or WT mice, the AUC_0–30min_ and C_max_ were not changed in Cyp3aXAV mice (Appendix A). Similar results were obtained for PIA (Appendix A). While not plotted separately, the PIA/EAI045 AUC ratios were also hardly changed between the three mouse strains. Overall, the modest changes we observed, as well as their unexpected direction, show that mouse Cyp3a and human CYP3A4 itself are unlikely to be major determinants of EAI045 or PIA pharmacokinetics in mice.

## 3. Discussion

In this study, we found that EAI045 is transported well in vitro by human ABCB1, moderately by canine ABCB1, and, at best, slightly by mouse Abcg2 and human ABCG2. In vivo mAbcb1a/1b had a significant effect on EAI045 plasma exposure and tissue distribution, with perhaps a slight contribution of mAbcg2. Although the absence of Abcb1a/1b and Abcg2 did not substantially increase the oral exposure of EAI045, the relative brain penetration of EAI045 was clearly increased (by 4.8-fold) when both Abcb1 and Abcg2 were ablated, and also by the absence of single Abcb1a/1b (3.9-fold increase). Similarly, the relative testis penetration of EAI045 was increased when both Abcb1a/1b and Abcg2 were absent (by 2.2-fold), and by the loss of single Abcb1a/1b (by 3.1-fold). These results demonstrate that Abcb1a/1b plays an important role in the BBB and BTB in restricting the brain and testis penetration of EAI045. We observed a significantly decreased SIWC%-to-plasma ratio of EAI045 in the small intestine with content in the mice lacking both Abcb1a/1b and Abcg2 (3.6-fold), but not in mice lacking only one of these systems. These findings support a role for both Abcb1a/1b and Abcg2 in the intestinal disposition of EAI045. As liver accumulation was not substantially changed by Abcb1a/1b and Abcg2 deficiency in the 30-min experiment, this suggests that Abcb1a/1b and Abcg2 primarily exert their effect by mediating direct efflux of EAI045 across the intestinal epithelium (thus also decreasing net intestinal uptake). In contrast, EAI045 distribution to kidney, spleen, and lung was not substantially affected by Abcb1a/1b and Abcg2 deficiency.

The brain accumulation of EAI045 was markedly increased in the absence of both Abcb1 and Abcg2, or in the absence of Abcb1 alone, but not in the absence of Abcg2. Since single Abcb1a/1b-deficient mice showed an increase in EAI045 brain accumulation almost equal to that in the combination knockout, Abcg2 makes, at best, a minor contribution in the BBB. Clearly Abcb1a/1b has a dominant role, and this also appears to apply to the BTB. These results are consistent with the in vitro transport data, showing that EAI045 is only a (very) weak substrate of mAbcg2 and hABCG2. Overall, it seems unlikely that hABCG2 will have a substantial effect on human EAI045 plasma exposure and tissue distribution; although, it should be kept in mind that protein expression of ABCG2 relative to ABCB1 in the human BBB is higher than in the mouse BBB [28]. This might result in a somewhat more prominent role of hABCG2 in the human BBB compared to mAbcg2 in the mouse BBB.

As brain metastases can easily occur in NSCLC, it is important to establish whether EAI045 has high intrinsic BBB permeability, and whether it interacts with ABCB1 and ABCG2 in the BBB. Up until now, the ability of EAI045 to penetrate the central nervous system (CNS) has not been well characterized, and there is very little publicly available information on the brain penetration of EAI045 in humans. In this study, we observed that in WT mice, the brain-to-plasma ratio was 0.10 at 30 min and 0.15 at 4 h after oral administration of EAI045 at 20 mg/kg. Therefore, compared to many other targeted anticancer drugs, EAI045 has an intermediate ability to cross the BBB even in WT mice. However, its brain penetration is still strongly limited by Abcb1a/1b, but not by Abcg2. The high in vitro polarized transport seen in ABCB1-overexpressing cells correlates well with the observed in vivo role of Abcb1a/1b in the BBB. This could mean that humans’ ABCB1 activity might also considerably restrict EAI045 brain accumulation.

Similar single and combined effects of Abcb1 and Abcg2 have been seen for other EGFR-targeting anticancer drugs, including afatinib, brigatinib, and osimertinib [29,30,31,32,33,34]. While these drugs and EAI045 are all substrates for ABCB1 and ABCG2 to varying extents, marked differences were seen in brain penetrance. Reported increases in brain penetration in *Abcb1a/1b;Abcg2^-/-^* compared to WT mice were as follows: brigatinib (38-fold), afatinib (28-fold), osimertinib (6.4-fold), and EAI045 (4.8-fold). Overall, the impact of Abcb1 and perhaps Abcg2 on brain accumulation of EAI045 was relatively modest compared to that seen for other EGFR inhibitors.

We note that the in vitro transport data with human ABCB1 (Figure 1C) and the in vivo data assessing mouse Abcb1a/1b function in the BBB (Figure 3) are qualitatively well in line. This is something we have consistently observed across at least 20 or more (experimental) drugs that we have tested over the years, with not a single exception (see e.g., [29,30,31,35,36], and many others). These results, therefore, strongly suggest that there is very extensive overlap in drug substrate specificity between human ABCB1 and mouse Abcb1a, which is the most prominent Abcb1 protein in the mouse BBB as well as small intestine.

We further observed that the dual ABCB1 and ABCG2 inhibitor elacridar could virtually completely reverse the impact of Abcb1a/1b and Abcg2 on restricting brain and testis penetration of EAI045, as well as limiting its net intestinal absorption. This suggests that elacridar coadministration could be considered to enhance the brain (and tumor) penetration of EAI045 in patients suffering from suitable EGFR mutant brain metastases. However, the 2.4-fold further increase in plasma AUC_0–30min_ of EAI045 seen in WT mice relative to *Abcb1a/1b;Abcg2^-/-^* mice upon elacridar coadministration shows an additional effect on its plasma kinetics independent of Abcb1a/1b and Abcg2 inhibition. This increase in oral exposure might present a toxicity risk of this combination treatment; although we did not find any acute externally visible toxicity in the mice during the experiment. Nonetheless, coadministration of elacridar with EAI045 in patients should only be considered with extreme caution, and it would be preferable if the alternative EAI045 clearance system affected by elacridar that appears to be present in mice would be first identified. Intriguingly, no additional changes in tissue distribution of EAI045 due to the elacridar treatment were observed in other tested tissues. On the one hand, this can be considered positive, as it reduces the risks of unexpected toxicities of elacridar co-treatment. On the other hand, this also does not provide any clues to the identity of the candidate EAI045 clearance system affected by elacridar. For instance, elacridar has been reported to inhibit human OATP1A2 [37]. However, if it would also inhibit mouse Oatp1a/1b and affect EAI045, this should have resulted in significantly decreased liver-to-plasma ratios of EAI045, which we did not find (Appendix A).

All in all, as human ABCB1 also transports EAI045, it might be possible to enhance the clinical effectiveness of EAI045 in treating (metastatic) brain tumors by inhibiting hABCB1, based on our in vitro and in vivo findings. Moreover, since tumor cells themselves also often express ABCB1, they can therefore be resistant to EAI045. Coadministration of ABCB1 inhibitors (such as elacridar) could thus improve the efficacy of EAI045 against these tumors. It may, therefore, be possible to further increase the effectiveness of EAI045 against brain metastases and perhaps even primary brain tumors by co-administration of an efficacious ABCB1/ABCG2 inhibitor. However, given the possible complications of elacridar with respect to EAI045 pharmacokinetics we observed, it might be worth considering alternative ABCB1/ABCG2 inhibitors.

Very little is known about the possible interaction of EAI045 with OATPs. We observed that Oatp1a/1b deficiency did not substantially change oral EAI045 plasma pharmacokinetics and liver distribution. Although there might possibly be a limited role of mOatp1a/1b transporters in EAI045 kidney accumulation, the changes in kidney-to-plasma ratios between WT and *Oatp1a/1b^-/-^* mice were modest (1.6-fold) (Appendix A). Moreover, the plasma and liver levels were not altered. Any wider in vivo impact of differences in OATP activity in patients on EAI045 pharmacokinetics is therefore likely to be small.

Not much is known about possible toxicity or drug–drug interactions mediated by the main EAI045 metabolites. There was little impact of Abcb1a/1b, Abcg2, and Oatp1a/1b on the plasma exposure and tissue penetration of the EAI045 hydrolyzed metabolite PIA. It is worth noting that in WT mice, the PIA/EAI045 AUC ratio was about 400% (data not shown), suggesting that a large amount of EAI045 was converted to PIA. However, the ratios did not differ significantly between strains, showing that Abcb1a/1b, Abcg2, or Oatp1a/1b do not have a marked role in, or impact on, the availability, metabolism, or elimination of PIA upon oral EAI045 administration.

We observed an increased plasma AUC_0–30min_ and a markedly decreased accumulation and tissue-to-plasma ratio of EAI045 glucuronide in the SIWC% when Abcb1a/1b and Abcg2 were absent. This suggests that Abcb1a/1b and Abcg2 are involved in the intestinal disposition of the glucuronide, possibly by mediating hepatobiliary excretion, or direct intestinal excretion, or both. Moreover, our data show that whereas the plasma AUC_0–30min_ of EAI045 was similar in WT and *Oatp1a/1b^-/-^* mice, the EAI045 glucuronide AUC_0–30min_ was 16.3-fold increased in *Oatp1a/1b^-/-^* mice (Figure 5E,F; Table 2). Accordingly, there was a dramatic decrease in liver concentration of EAI045-glucuronide, from 1.8% of the administered EAI045 dose in WT mice to less than 0.8% in the Oatp1a/1b knockout mice (Appendix A), with a 22.6-fold decreased liver-to-plasma ratio (Figure 6F–H) in *Oatp1a/1b^-/-^* mice relative to WT mice. This clearly demonstrates that Oatp1a/1b is markedly involved in the transport of EAI045 glucuronide from blood into the liver. SIWC levels of EAI045 glucuronide were also strongly decreased in the Oatp1a/1b knockout mice. This was most likely because of relatively reduced hepatobiliary excretion as a consequence of the reduced liver levels of the glucuronide.

Our 24-h experiments on oral EAI045 in *Cyp3a^-/-^* and Cyp3aXAV mice suggest there may, at best, be a very limited impact of mouse Cyp3a or human CYP3A4 on EAI045 oral exposure (Appendix A). Furthermore, the conversion of EAI045 to PIA is unlikely to be primarily mediated by CYP3A, given minimal changes in PIA/EAI045 ratios between the strains. However, the slightly decreased plasma AUC_0–30min_ in *Cyp3a^-/-^* mice (1.6-fold) suggests that there might be other system(s) that can metabolize EAI045, perhaps Cyp2c enzymes, which are known to be compensatory upregulated in *Cyp3a^-/-^* mice [38].

## 4. Materials and Methods

### 4.1. Chemicals

EAI045 was obtained from Carbosynth (Berkshire, UK). Elacridar HCl and zosuquidar were purchased from Sequoia Research Products (Pangbourne, UK). Ko143 was from Tocris Bioscience (Bristol, UK). Glucose water (5%, *w*/*v*) was supplied by B. Braun Medical Supplies (Melsungen, Germany). Bovine Serum Albumin (BSA) Fraction V was obtained from Roche Diagnostics GmbH (Mannheim, Germany). Isoflurane was from Pharmachemie (Haarlem, The Netherlands), and heparin (5000 IU mL^−1^) was purchased from Leo Pharma (Breda, The Netherlands). Other chemicals used in the EAI045 assays were described before [35,36], and all other chemicals and reagents were obtained from Sigma-Aldrich (Steinheim, Germany).

### 4.2. Transport Assays

Madin–Darby Canine Kidney (MDCK-II) cells (ECACC 00062107) stably transduced with human (h) ABCB1, hABCG2, or mouse (m) Abcg2 cDNA were generated in our institute between 1998 and 2005. Transepithelial transport assays were performed on microporous polycarbonate membrane filters (3.0 mm pore size, 12 mm diameter, Transwell 3402, Corning Incorporated, Kennebunk, ME). In short, cells were allowed to grow an intact monolayer in 3 days, which was monitored with transepithelial electrical resistance (TEER) measurements, both before and after the transport phase. On the day of the experiment, appropriate cells were pre-incubated (1 h) with 5 µM zosuquidar (ABCB1 inhibitor) and/or 5 µM Ko143 (ABCG2/Abcg2 inhibitor) in both compartments. The transport phase was then initiated (t = 0) by replacing the medium in apical or basolateral compartments with fresh Dulbecco’s Modified Eagle’s medium (DMEM medium) including 10% (*v*/*v*) fetal bovine serum (FBS) and 5 μM EAI045, as well as the appropriate inhibitor(s). Cells were kept at 37 °C in 5% (*v*/*v*) CO_2_ during the experiment. Samples of 50 μL were taken from the acceptor compartment at 1, 2, 4, and 8 h, and stored at −30 °C until LC-MS/MS analysis. Active transport was expressed using the transport ratio (r), which is defined as the amount of apically directed drug transport divided by basolaterally directed drug translocation at a defined time point. Proper functionality of these cell lines was monitored by continual testing with a variety of drug substrates.

### 4.3. Animals

Mice (*Mus musculus*) were bred in the Netherlands Cancer Institute and housed and handled according to institutional guidelines complying with Dutch and EU legislation. Animals used were WT, *Abcb1a/1b^-/-^* [39], *Abcg2^-/-^* [40], *Abcb1a/1b;Abcg2^-/-^* [41], *Cyp3a^-/-^* [42], and *Oatp1a/1b^-/-^* [43] mice of a >99% FVB/N strain background. Homozygous CYP3A4 humanized transgenic mice (Cyp3aXAV) were generated by cross-breeding of transgenic mice with stable human CYP3A4 expression in the liver or intestine, respectively, in a *Cyp3a^-/-^* background [42]. Proper functionality of the knockout and transgenic strains was monitored by regular testing with a variety of drug substrates. All the mice used were between 10 and 15 weeks of age. Animals were kept in a temperature-controlled environment with a 12 h light/12 h dark cycle and received a standard diet (Transbreed, SDS Diets, Technilab-BMI, The Netherlands) and acidified water ad libitum. Depending on the type of experiment, 5 or 6 mice were tested in each experimental group. All experimental animal protocols, including power calculations, were ethically assessed and designed under the nationally approved DEC/CCD project AVD301002016595 (approval date, 10 August 2016) and evaluated and approved by the institutional animal care and use committee (IACUC) of the Netherlands Cancer Institute.

### 4.4. Working Solutions

EAI045 was dissolved in dimethyl sulfoxide (DMSO) at a concentration of 4 mg/mL, and then diluted with a mixture of polysorbate 80/ethanol (1:1, *v*/*v*), and further diluted with 5% (*w*/*v*) glucose in water to yield a concentration of 2 mg/mL. Final concentrations for DMSO, polysorbate 80, ethanol, and glucose in the dosing solution were 5%, 2.5%, 2.5%, and 4.5% (*v*/*v*/*v*/*w*), respectively. Elacridar hydrochloride was dissolved in DMSO at a concentration of 53 mg/mL, then diluted with a mixture of polysorbate 80, ethanol, and water [20:13:67, (*v*/*v*/*v*)] to yield a concentration of 5 mg/mL elacridar. The final oral dose of elacridar was 50 mg/kg body weight. All dosing solutions were prepared freshly on the day of experiment.

### 4.5. Plasma and Tissue Pharmacokinetics of EAI045

To minimize variation in absorption, mice were fasted for about 3 h before EAI045 was administered by gavage into the stomach (n = 5–6), using a blunt-ended needle. EAI045 was administered orally at dose of 20 mg/kg. Elacridar was administered orally at a dose of 50 mg/kg, 3 h prior to oral EAI045 administration. For the 24- or 4-h experiments, tail vein blood samplings were performed at 5 min, 30 min, 1, 2, 4, and 8 h, or 5 min, 15 min, 30 min, 1, and 2 h time points after oral administration of EAI045, respectively, using microvettes containing heparin. For the 30 min experiments, blood samplings were performed at 5, 10, and 20 min from the tail vein. At the last time point (24 h, 4 h, or 30 min), mice were anesthetized with isoflurane and blood was collected by cardiac puncture. Then, mice were sacrificed by cervical dislocation and brain, liver, spleen, kidneys, lung, small intestine with content (SIWC), and testis were rapidly removed and weighed. Tissues were homogenized on ice in appropriate volumes of 2% (*w*/*v*) BSA. Blood samples were centrifuged at 9000× *g* for 6 min at 4 °C immediately after collection. All samples were stored at −30 °C until analysis.

### 4.6. LC-MS/MS Analysis

Concentrations of EAI045 (Appendix A) and its hydrolyzed metabolite, 5-fluoro-2-hydroxyphenyl)(1-oxo-1,3-dihydro-2H-isoindol-2-yl)acetic acid, abbreviated PIA for (phenyl-(iso)indole-acetic acid) (Appendix A), and EAI045 glucuronide response relative to EAI045 in cell culture medium, plasma, and tissue homogenates were analyzed with a liquid-chromatography tandem mass spectrometric (LC-MS/MS) assay [44]. Due to the absence of an authentic standard, EAI045 glucuronide could only be semi-quantified, but this semi-quantification does allow relative concentration comparisons to be made within one tissue or matrix. The signal response of EAI045 glucuronide was assumed to be equivalent to that of the parental EAI045. Therefore, all “ng” units used for EAI045 glucuronide should be considered as arbitrary mass units, which could well represent quite different absolute amounts of compound from the validated parent EAI045 ng units.

### 4.7. Pharmacokinetic and Statistical Calculations

Pharmacokinetic parameters were calculated using the software GraphPad Prism7 (GraphPad Software, La Jolla, CA, USA). The area under the plasma concentration-time curve (AUC) was calculated using the trapezoidal rule with the Microsoft Excel plug in PKsolver [45], without extrapolating to infinity. The peak plasma concentration (C_max_) and the time to reach C_max_ (t_max_) were estimated from the original data. Ordinary one-way analysis of variance (ANOVA) was used to determine significance of differences between groups, after which post hoc tests with Tukey correction were performed for comparison between individual groups. The two-sided unpaired Student’s *t*-test was used when treatments or differences between two groups were compared. When variances were not homogeneously distributed, data were log-transformed before statistical tests were applied. Differences were considered statistically significant when *p* < 0.05. Data are presented as mean ± SD.

## 5. Conclusions

In summary, the most striking finding of this study is that Abcb1a/1b was responsible mostly for limiting the EAI045 brain and testis penetration, but not its oral exposure. Moreover, co-administration of elacridar could substantially enhance the oral availability and brain penetration of oral EAI045. Additionally, we demonstrated that Oatp1a/1b is markedly involved in the transport of EAI045 glucuronide from the blood into the liver. Lastly, both mCyp3a and human CYP3A4 seem to have little, if any, direct effect on the in vivo oral exposure of EAI045; although we cannot exclude that there are meaningful alterations in other EAI045-detoxifying systems. Collectively, our findings can be considered positive for the clinical application of EAI045, as they indicate that CYP3A- or ABC/OATP transporter-related drug–drug interactions or genetic polymorphisms would probably be of minor concern.

## Figures and Tables

**Figure 1 pharmaceuticals-15-01124-f001:**
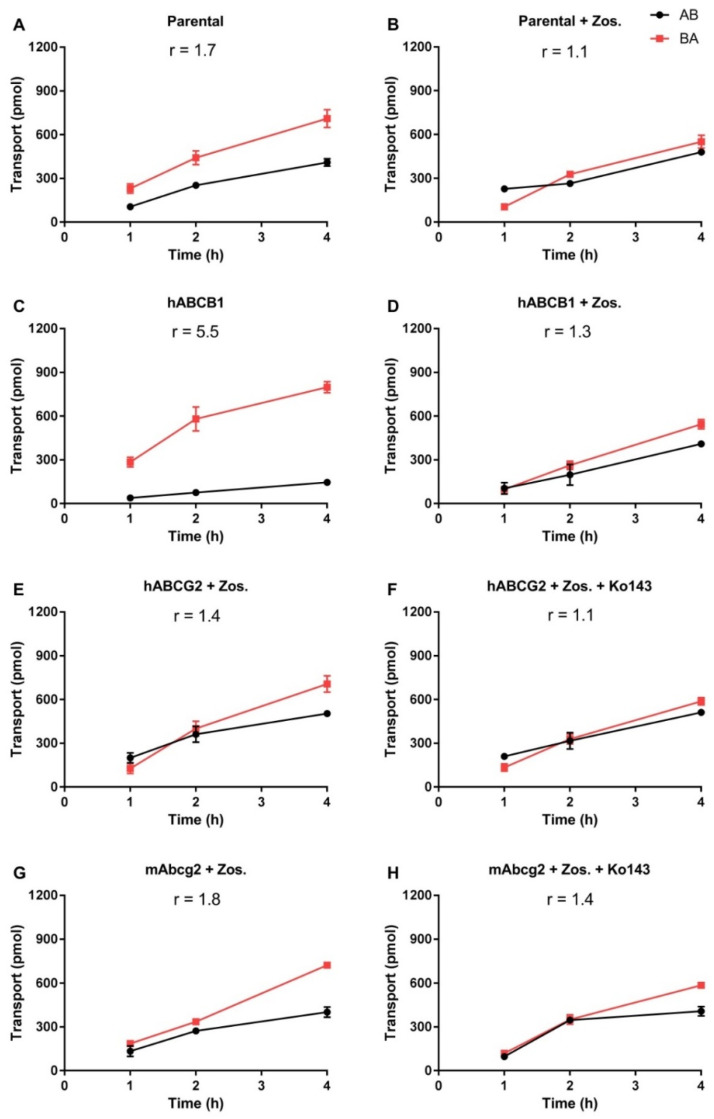
In vitro transepithelial transport of EAI045 (5 μM) using MCDK-II parental cells (**A**,**B**) or transduced with human ABCB1 (**C**,**D**), human ABCG2 (**E**,**F**), or murine Abcg2 (**G**,**H**) cDNA. At t = 0 EAI045 was added in the donor compartment (either apical or basolateral). At 1, 2, and 4 h EAI045 was quantified in the acceptor compartment and the cumulative total transported amount per well was plotted (n = 3). Zosuquidar (5 μM) was added to inhibit the human and/or the endogenous ABCB1, while Ko143 (5 μM) was added to inhibit the human or murine ABCG2. r, relative transport ratio. BA (■), translocation from the basolateral to the apical compartment; AB (●), translocation from the apical to the basolateral compartment. Points, mean; bars, SD.

**Figure 2 pharmaceuticals-15-01124-f002:**
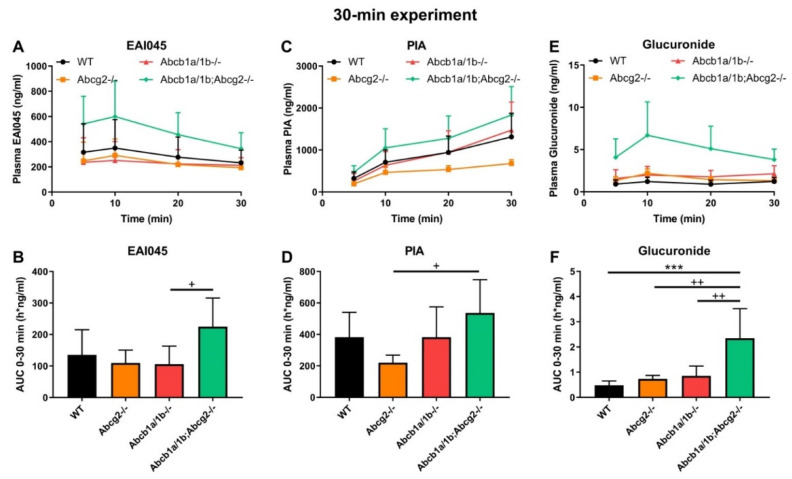
Plasma concentration-time curves and AUC_0–30min_ of EAI045 (**A**,**B**), PIA (**C**,**D**), and EAI045 glucuronide (**E**,**F**) in male WT, *Abcb1a/1b^-/-^*, *Abcg2^-/-^* and *Abcb1a/1b;Abcg2^-/-^* mice 30 min after oral administration of 20 mg/kg EAI045. Data are given as mean ± S.D. (n = 6–7). ***, *p* < 0.001 compared to WT mice. ^+^, *p* < 0.05; ^++^, *p* < 0.01 compared to *Abcb1a/1b;Abcg2^-/-^* mice.

**Figure 3 pharmaceuticals-15-01124-f003:**
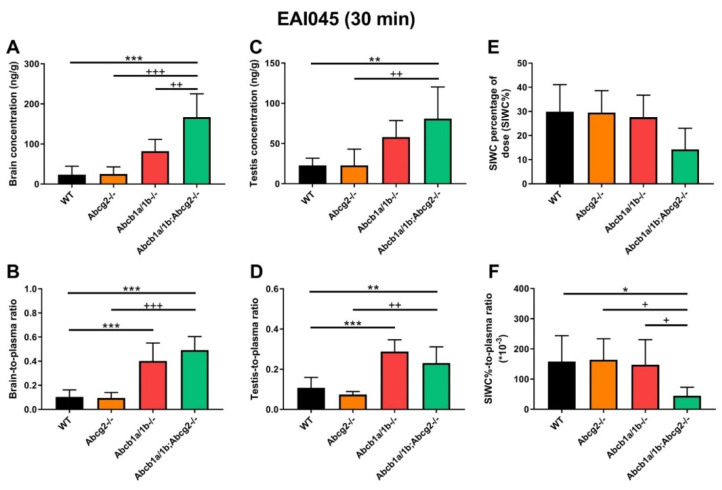
Organ concentration (**A**,**C**,**E**), and organ-to-plasma ratio (**B**,**D**,**F**) of EAI045 in male WT, *Abcb1a/1b^-/-^*, *Abcg2^-/-^* and *Abcb1a/1b;Abcg2^-/-^* mice 30 min after oral administration of 20 mg/kg EAI045. Data are given as mean ± S.D. (n = 6–7). *, *p* < 0.05; **, *p* < 0.01; ***, *p* < 0.001 compared to WT mice. ^+^, *p* < 0.05; ^++^, *p* < 0.01; ^+++^, *p* < 0.001 compared to *Abcb1a/1b;Abcg2^-/-^* mice.

**Figure 4 pharmaceuticals-15-01124-f004:**
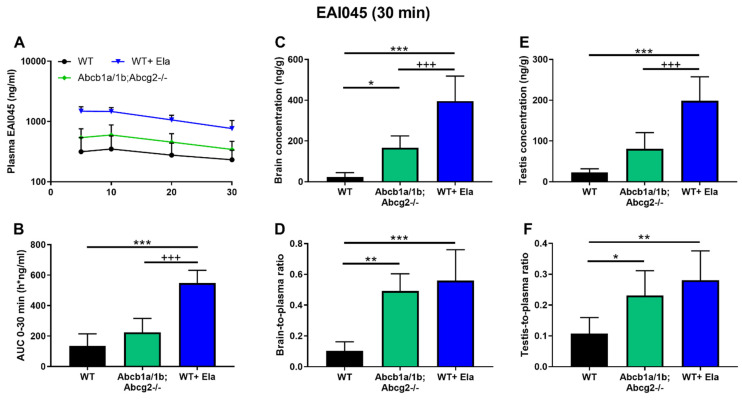
Plasma concentration-time curves (**A**), plasma AUC_0–30min_ (**B**), brain and testis concentration (**C**,**E**), and brain- and testis-to-plasma ratio of EAI045 (**D**,**F**) in male WT and *Abcb1a/1b;Abcg2^-/-^* mice over 30 min after oral administration of 20 mg/kg EAI045 with or without co-administration of elacridar (Ela). Data are given as mean ± S.D. (n = 6–7). *, *p* < 0.05; **, *p* < 0.01; ***, *p* < 0.001 compared to WT mice. ^+++^, *p* < 0.001 compared to *Abcb1a/1b;Abcg2^-/-^* mice.

**Figure 5 pharmaceuticals-15-01124-f005:**
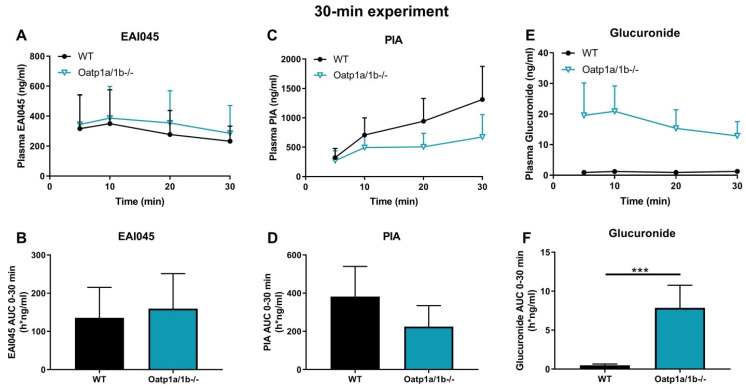
Plasma concentration-time curves and AUC_0–30min_ of EAI045 (**A**,**B**), PIA (**C**,**D**), and EAI045 glucuronide (**E**,**F**) in male WT and *Oatp1a/1b^-/-^* mice 30 min after oral administration of 20 mg/kg EAI045. Data are given as mean ± S.D. (n = 6–7). ***, *p* < 0.001 compared to WT mice.

**Figure 6 pharmaceuticals-15-01124-f006:**
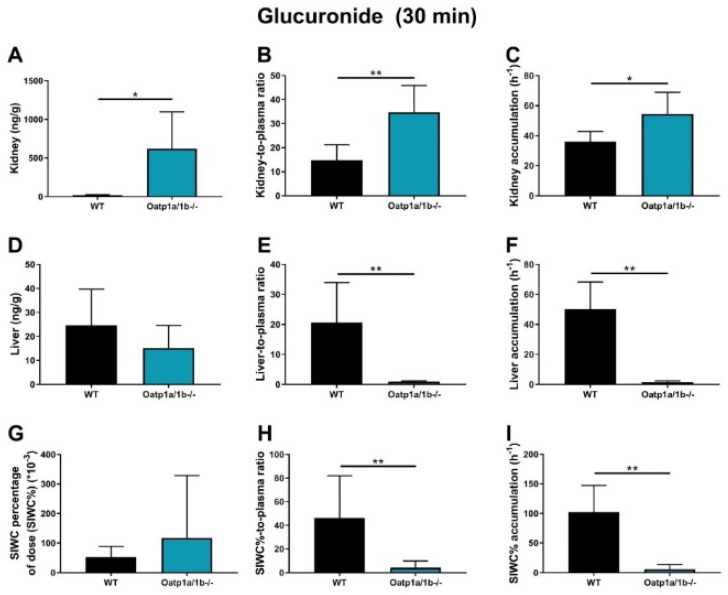
EAI045−glucuronide response relative to EAI045 concentration in each tissue and matrix (**A**,**D**,**G**), response-to-plasma ratio (**B**,**E**,**H**) and response accumulation (**C**,**F**,**I**) in male WT and *Oatp1a/1b^-/-^* mice 30 min after oral administration of 20 mg/kg EAI045. Data are given as mean ± S.D. (n = 6−7). *, *p* < 0.05; **, *p* < 0.01 compared to WT mice.

**Table 1 pharmaceuticals-15-01124-t001:** Plasma and tissue pharmacokinetic parameters of EAI045 in male WT, *Abcb1a/1b^-/-^*, *Abcg2^-/-^* and *Abcb1a/1b;Abcg2^-/-^* mice over 30 min after oral administration of 20 mg/kg EAI045 without or with inhibitor elacridar (Ela, only for WT mice).

Parameter (EAI045)	Genotype				
	WT	*Abcg2^-/-^*	*Abcb1a/1b^-/-^*	*Abcb1a/1b;Abcg2^-/-^*	WT + Ela
Plasma AUC_0–30min_ (h*ng/mL)	136 ± 80	110 ± 41	106 ± 57 ^+^	225 ± 91	549 ± 82 ***^. +++^
Fold change AUC	1.0	0.81	0.78	1.7	4.0
t_max_ (min)	5–30	10–30	10–30	10–30	10–30
C_max_ (ng/mL)	354 ± 225	310 ± 114	201 ± 79	603 ± 277	1512 ± 260
C_brain_ (ng/g)	23.3 ± 21.3	25.2 ± 17.6 ^+++^	82.0 ± 29.7 ^++^	166.9 ± 58.4 ***	395.9 ± 123.1 ***^. +++^
Fold change C_brain_	1.0	1.1	3.5	7.2	17.0
K_p.brain_	0.103 ± 0.059	0.094 ± 0.046 ^+++^	0.401 ± 0.149 ***	0.492 ± 0.112 ***	0.560 ± 0.200 ***
Fold change K_p.brain_	1.0	0.92	3.9	4.8	5.4
C_testis_ (ng/g)	22.8 ± 8.9	22.8 ± 20.2 ^++^	70.9 ± 36.2	80.9 ± 39.4 **	198.7 ± 58.9 ***^. +++^
Fold change C_testis_	1.0	1.0	3.1	3.5	8.7
K_p.testis_	0.107 ± 0.052	0.074 ± 0.015 ^++^	0.329 ± 0.114 ***	0.231 ± 0.080 **	0.280 ± 0.096 **
Fold change K_p.testis_	1.0	0.69	3.1	2.2	2.6
SIWC%	29.9 ± 11.3	29.5 ± 9.1	27.6 ± 9.2	14.2 ± 8.8	4.00 ± 2.57 ***
Fold change SIWC%	1.0	0.99	0.92	0.48 (2.1-fold)	0.13 (7.5-fold)
K_p.SIWC%_	158 ± 86	164 ± 70 ^+^	148 ± 82 ^+^	44.4 ± 28.3 *	5.13 ± 2.63 ***
Fold change K_p.SIWC%_	1.0	1.04	0.94	0.28 (3.6)	0.032 (30.8)

Data are given as mean ± S.D. (n = 6–7). AUC_0–30min_, area under the plasma concentration-time curve; C_max_, maximum concentration in plasma; t_max_, time point (min) of maximum plasma concentration; C_tissue_, tissue concentration; K_p.tissue_, tissue-to-plasma ratio. SIWC, small intestine with content; SIWC%, Drug percentage of dose in SIWC, expressed as total drug (ng) in SIWC divided by total drug administered (ng). *, *p* < 0.05; **, *p* < 0.01; ***, *p* < 0.001 compared to WT mice. ^+^, *p* < 0.05; ^++^, *p* < 0.01; ^+++^, *p* < 0.001 compared to *Abcb1a/1b;Abcg2^-/-^* mice.

**Table 2 pharmaceuticals-15-01124-t002:** Plasma and tissue pharmacokinetic parameters of EAI045 glucuronide in male WT and *Oatp1a/1b^-/-^* mice over 30 min after oral administration of 20 mg/kg EAI045.

Parameter (Glucuronide)	Genotype	
	WT	*Oatp1a/1b^-/-^*
Plasma AUC_0–30min_ (h*ng/mL)	0.483 ± 0.168	7.86 ± 2.91 ***
Fold change AUC	1.0	16
t_max_ (min)	5−30	5–30
C_max_ (ng/mL)	1.53 ± 0.55	22.2 ± 8.9
Kidney	17.6 ± 7.3	622 ± 475 *
Fold change kidney	1.0	35
K_p.Kidney_	14.8 ± 6.5	34.8 ± 11.1 **
Fold change K_p.Kidney_	1.0	2.4
P_Kidney_ (*10^−6^h^−1^)	36.0 ± 6.8	54.6 ± 14.3 *
Fold change P_Kidney_	1.0	1.5
Liver	24.7 ± 15.0	15.2 ± 9.4
Fold change liver	1.0	0.62
K_p.Liver_	20.7 ± 13.3	0.915 ± 0.315 **
Fold change K_p.Liver_	1.0	0.044
P_Liver_ (*10^−6^h^−1^)	50.3 ± 18.0	1.54 ± 0.87 **
Fold change P_Liver_	1.0	0.031
SIWC%	52.3 ± 36.1	117 ± 211
Fold change SIWC%	1.0	2.2
K_p.SIWC%_	46.2 ± 35.7	4.08 ± 5.72 **
Fold change K_p.SIWC%_	1.0	0.088
P_SIWC%_ (*10^−6^h^−1^)	102 ± 45	5.80 ± 7.84 **
Fold change P_SIWC%_	1.0	0.057

Data are given as mean ± S.D. (n = 6–7). AUC_0–30min_, area under the plasma concentration-time curve; C_max_, maximum concentration in plasma; t_max_, time point (min) of maximum plasma concentration; C_tissue_, tissue concentration; K_p.tissue_, tissue-to-plasma ratio. P_Kidney_ and P_Liver_ were calculated by dividing the tissue concentrations at 30 min by the plasma AUC. SIWC, small intestine tissue with content; SIWC%, C: Drug percentage of dose in SIWC, which was expressed as total drug (ng) in SIWC divided by total drug administered (ng). *, *p* < 0.05; **, *p* < 0.01; ***, *p* < 0.001 compared to WT mice.

## Data Availability

Data is contained within the article and Appendix A.

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
