# Peer review of "P-Glycoprotein (MDR1/ABCB1) Restricts Brain Accumulation of the Novel EGFR Inhibitor EAI045 and Oral Elacridar Coadministration Enhances Its Brain Accumulation and Oral Exposure"

_pharmaceuticals, 2022, doi:10.3390/ph15091124_

Round 1

Reviewer 1 Report

In this manuscript, Wang and colleagues describe the potential impact of various transporters on the disposition of the novel fourth-generation allosteric epidermal growth factor receptor TKI, EAI045.  Using MDCK-II cells transfected with hABCB1, hABCG2 and mAbcg2 as well as the transport inhibitors Zosuquidar and Ko143, they demonstrate a major role for hABCB1 and minor roles for mAbcg2 and hABCG2 and potentially canine ABCB1 in transporting EAI045.  In more complex in vivo studies, they evaluate EAI045 PK in mice deficient in murine Abcb1a/1b, Abcg2, the combination of both transporters, OATp1a/1b, or Cyp3a as well as mice transgenic for human Cyp3A4 in a Cyp3a-/- background (Cyp3aXAV).  They also utilize the transport inhibitor elacridar as well as explore the disposition of two metabolites of EAI045, the hydrolysis product PIA and a glucuronidated metabolite with unknown structure.  Overall the data suggest that 1) ABCB1 substantially limits EAI045 brain (and testes) accumulation; 2) the metabolite PIA is unaffected by any of the models evaluated; 3) the glucuronidated metabolite seems most affected by OATp1 deficiency but the contribution of this metabolite to overall parental drug disposition may be limited due to low concentrations (although unclear how this metabolite was quantitated since the structure is unknown); and 4) elacridar increased brain accumulation and overall plasma exposure but did so more significantly than KO of Abcb1/1b and Abcg2, suggesting it may inhibit more than just these two transporters.

While the data were interesting, there were so many figures presented in so many different ways that it was difficult to keep track of the major findings.  There are potentially a number of things that could be done to simplify the presentation.

1) The authors could consider not showing data for the PIA metabolite and the impact of Cyp3a and instead just describe the lack of effect in the text.

2) All data were gathered after oral administration of EAI045.  To assess the impact of transporters on tissue distribution separate from the impact of transporters on intestinal absorption, it would be helpful to evaluate tissue distribution after IV administration and then compare those effects to what was observed after oral administration.  Including elacridar in this experiment could potentially help sort out where it is having the bigger impact (absorption vs tissue distribution).

3) Given the more major impact in vivo of mAbcb1a/1b, it would be helpful if there were a comparable MDCK-II assay utilizing these transporters so that a more direct comparison could be made between these transporters and mAbcg2 as well as to the effect of  human ABCB1.

4) Please describe how EAI045 glucuronide concentrations are calculated using ng/ml or ng/g for Fig 5E, Fig 6 A, D, Suppl Fig 6B, F without a standard.  Please also describe in more detail how accumulation for the glucuronide is calculated (Fig 6, Suppl Fig 6) and include these calculations in the Methods.

5) Please describe Pkidney and PLiver and calculation thereof as used in Table 2.

6) If there is anything known of effects of elacridar on other transporters, it might be helpful to include this information in the discussion.

7) The legend does not match the figure headings in Suppl Fig 2.  Please clarify.

Reviewer 2 Report

The authors made great efforts to demonstrate that EAI045 is transported in vitro effectively by human ABCB1, somewhat by canine ABCB1, and at best by mouse Abgg2 and human ABCG2. They also, study the in vivo, mAbcb1a/1b substantial effect on the pharmacokinetics and tissue distribution of EAI045, with mAbg2.

The following comments and suggestions may improve the manuscript quality:

-          Generally, please check the manuscript for typo mistakes and spaces, L245 “latter ??”. L 308 paraphrase sentences such that in the abstract L23, “EAI045 brain-to-plasma ratios were increased by 3.9-fold in Abcb1a/1b-/- and 4.8-fold in Abcb1a/1b; Abcg2-/- mice compared to wild-type mice, but not in single Abcg2-/- mice”.

-          The same case at L 137, we next measure…, correct please, the same in L 170.

-          L 84, we determined…please change to, we aimed to determine….., check for similar situations throughout the manuscript.

-          Supplemental figures are missed, the authors mentioned 10 supplemental figures, the last one in line 386, this file is not founded as a file or URL to download it, please upload it, many paragraphs in the results and discussion sections mentioned supplemental figures missed, e.g L 247 Figures 5 C-P ??? L 338 Figures 8A-H, K-N,….

-          Figure 1, shift it to section 2.1.

In figure 2 in the capture, remove *, P < 0.05; **, P < 0.01; as there are no results fitted to those significant intervals, do that check throughout all figures.

-          Figure 3 E, check p- values were missed.

-          Line 263, Due to absence…., please mention that as a limitation issue at the end of the manuscript.

-          2.7. change to Assessment of the in vivo role…..

-          Lack of references and references update should be revised as there is only about 5 ref. between 2017-2019. (ref, 27,37-39). Please update references.

Round 2

Reviewer 1 Report

The textual changes made by the authors have improved the readability of the manuscript and simplified the findings, such that the major conclusions are more readily apparent.  The reduction in supplemental figures has also helped.

The emphasis on tissue/plasma levels has helped clarify the authors' contention that IV administration is not necessary for clarification of the role of Abcb1a/1b and Abcg2 on EA1045 PK.  The role of elacridar is less clear but sorting that out would likely require significant effort beyond the scope of this manuscript.  However, this is something that would be ultimately important in order to consider co-administration as the authors rightly point out but not needed here for publication.

Two minor points to consider

1) it would be helpful if the authors included in the methods section their assumption that the signal response given by the glucuronide was equivalent to the signal response given by EA1045 itself in their semi-quantitative measurement of glucuronide levels in addition to the statement they have already added to make more clear how the quantitation was done.  Others in the field may rely on their measurements, even if the authors here are only wanting to make relative comparisons.  

2) Given the author's pre-eminence in the transporter field, it might be helpful if they could include a statement in the results or discussion (or if possible include a prior reference) that emphasizes their observations that hABCB1 and murine Abcb1a/1b show largely overlapping substrate specificity as this may be of interest to the field at large. 

Reviewer 2 Report

Many thanks for following the comments and enhancing the manuscript quality, still minor issues:

1- Check spaces and minor spelling errors after removal of changes by tracking.

2- Regarding supplementary figures, the authors replace them with (data not shown)????...please add figures legend at their right position and the reader will refer to them in the supplementary file. example: line 271...Supp Fig 5C...apply that to all supplementary figures

3- In tables, remove dark areas. 

4- Lines 261-276 can be separated in the Conclusions section at the end of your manuscript. (5. Conclusions. This section is not mandatory but can be added to the manuscript if the discussion is unusually long or complex)
